# Distribution of Biodiversity of Wild Beet Species (Genus *Beta* L.) in Armenia under Ongoing Climate Change Conditions

**DOI:** 10.3390/plants11192502

**Published:** 2022-09-24

**Authors:** Anna Avetisyan, Tatevik Aloyan, Amalya Iskandaryan, Margarita Harutyunyan, Laura Jaakola, Andreas Melikyan

**Affiliations:** 1Scientific Center of Agrobiotechnology, ANAU Armenian National Agrarian University, Teryan 74, Yerevan 0009, Armenia; 2Department of Arctic and Marine Biology, UiT The Arctic University of Norway, NO-9037 Tromso, Norway; 3Norwegian Institute of Bioeconomy Research, NO-1431 Ås, Norway

**Keywords:** wild relative, phylogenetic tree, genetic resources, species biogeography, ecosystems, species conservation, herbarium, *Amaranthaceae*, *Corollinae*, *Beta x intermedium* Aloyan

## Abstract

The reported annual temperature increase and significant precipitation drop in Armenia impact the country’s ecosystems and biodiversity. The present study surveyed the geographical distribution of the local wild beet species under the ongoing climate change conditions. We showed that *B. lomatogona*, *B. corolliflora* and *B. macrorhiza* are sensitive to climate change and were affected to various degrees, depending on their location. The most affected species was *B. lomatogona*, which is at the verge of extinction. Migration for ca. 90 and 200–300 m up the mountain belt was recorded for *B. lomatogona* and *B. macrorhiza*, respectively. *B. corolliflora* was found at 100–150 m lower altitudes than in the 1980s. A general reduction in the beet’s population size in the native habitats was observed, with an increased number of plants within the populations, recorded for *B. corolliflora* and *B. macrorhiza.* A new natural hybrid *Beta x intermedium* Aloyan between *B. corolliflora* and *B. macrorhiza* was described and confirmed using chloroplast DNA trnL-trnF intergenic spacer (LF) and partially sequenced alcohol dehydrogenase (*adh*) of nuclear DNA. An overview of the wild beets reported in Armenia with the taxonomic background, morphological features, and distribution is provided. Conservation measures for preservation of these genetic resources are presented.

## 1. Introduction

The Republic of Armenia is a small landlocked mountainous country in the South Caucasus at the northern end of the Armenian Highland. It has an elevation ranging from 375 m above sea level (a.s.l.) in the Arax river valley to 4090 m a.s.l. at Mt. Aragats. About 85% of the Armenia’s area is at an altitude above 1000 m; and some 51% of its area is above 2000 m. The country’s complex relief, geological history, and local climatic conditions contributed to the formation of several phytogeographic regions with diverse ecosystems and rich and unique biodiversity. Ten landscape zones in Armenia shape all the main natural ecosystems of the Caucasus, except moist subtropical ecosystems. There is also a number of intrazonal ecosystems (wetlands, rocks, and screes), which are present almost in all altitudinal zones (Appendix A, [1]). The number of vascular plant species described in the country is ca. 3800, of which the number of endemic species is 144, comprising 3.8% of total flora [1,2]. Wild relatives of numerous cultivated plants still occur in the territory of Armenia, making Armenia a globally significant center of origin of agrobiodiversity. Besides their essential role in ecosystems, crop wild relatives provide a specific source of genetic material for creating new varieties and improving quality of crops, affecting their productivity and adaptation [3,4,5,6,7]. 

Armenia’s climate can be described as highland continental, with large spatial and seasonal variations. Because of the country’s complex landscape (Appendix A) there is a climate range from arid to sub-tropical and to cold, high mountains. Relatively high temperatures can be seen in Ararat Valley (south-west), the north-eastern and south-eastern valley regions of Armenia. The more mountainous regions experience lower average temperatures, with sub-zero average annual temperatures at altitudes above 2500 m [8,9]. The strong annual cycle of temperature well-defines differences for all four seasons (Appendix A, [8]). The 1991–2020 average winter and summer temperatures are −5.57 and 19.65 °C, respectively, with transitional 7.04 and 9.45 °C mean values in spring and autumn [8]. The average annual precipitation is low at 526 mm [8]. Precipitation intensity is greater in Armenia’s high-altitude regions with May and June as the wettest months. Armenia’s highest peaks may receive up to 1000 mm of annual precipitation while precipitation can be as low as 200 mm in the western plains [8]. Great spatial and temporal variability in both temperature and precipitation may be explained by the country’s geographic location, topography of the area, and the broad characteristics of the global circulation and regional climate factors (e.g., latitude, orography, oceanic, and continental influences) [10,11,12]. 

Armenia’s communication to the Convention on Climate Change has presented the best available evidence on the country’s climate change, reporting that as a mountainous country with a dry climate Armenia is vulnerable to global climate change. From 1929 to 2016, Armenia’s annual mean temperature increased by 1.23 °C, and the annual average precipitation decreased by about 9% from 1935 to 2016. Most of this decrease occurred in the past two decades, a time that was characterized by an increased frequency and intensity of extreme weather events and natural hazards such as droughts, heat waves, hailstorms, and forest fires [13]. 

These changes cause alterations in phenological indicators of species, species composition in communities, and increased risk of elimination of native species including endemics [14,15]. Armenia’s 2019 Sixth National Report to the Convention on Biological Diversity reported that forest areas in Armenia have been significantly reduced; and while semi-desert and steppe vegetation zones have expanded, the alpine vegetation zone has shrunk. Moreover, semi-deserts have experienced a vertical shift of the limits of the flora distribution up the mountain profile by 50 m and show signs of more active soil erosion and desertification [2].

Predictions of climate change suggest that Armenia could experience warming, strongly biased towards the summer months at levels significantly above the global average, with a potential increase of 4–6 °C under the highest emission scenarios by the end of the century. There is a projected decline in atmospheric precipitation, however, the precise assessments of future precipitation levels are uncertain due to the large number of affecting factors. While vulnerability for water resources in Armenia varies, under a ‘worst-case scenario’ the average decrease in river flow is estimated at 39% by 2100 [9,13,16]. 

Such prognosis stresses the necessity of closer attention to monitoring of the distribution of wild species and outlining the measures for vulnerability assessment for biodiversity conservation under the changing climate conditions. 

The flora of Armenia provides a great variety of agriculturally valuable plants including medicinal, edible, nonfood crop, and fodder plants. Beets are agriculturally important vegetables, used for edible leaves and roots, for sugar production, and cultivated also as a livestock fodder crop. A high content of biologically active substances with health-promoting properties, such as betalains, polyphenols, folates, minerals and vitamins, contribute to anti-inflammatory, antioxidant, anticancerogenic, anti-ischemic, and hepatoprotective activities of these plants [17,18]. Beetroot juice and extracts also serve as a traditional medicine to treat constipation, gut and joint pain, dandruff, and may be used as food colorants and additives to cosmetics [17,19,20]. Beets were domesticated in the ancient Middle East some 8000 years ago primarily for their greens [21,22]. By the Roman era, it is thought that they were cultivated for their roots as well and were used medicinally as a treatment for a variety of conditions [17,21]. Beets were mentioned also in the Armenian manuscripts. Mkhitar Heratsi, the 12th-century Armenian physician and founder of the medieval Armenian medicine, mentioned beet root in his “Consolation of Fevers”, where he suggested that patients use beet root as a remedy for a number of illnesses [19]. Describing beets in the encyclopedic “*Haybusak or Armenian Botany*” work on the flora of the Armenian highlands Levon (Ghewond) Alishan in 1895 noted diverse local beet populations in Armenia [23].

Cultivated varieties of beet (*Beta vulgaris*) together with their wild relatives are included into the genus *Beta* within the *Amaranthaceae* family (former *Chenopodiaceae*), which was included in a global priority conservation list of 92 crop wild relative genera in 2013 [24]. The genus is divided into two sections: *Beta* sect. *Beta*, including, *B. macrocarpa* Guss., *B. patula* Aiton and *B. vulgaris* L. species and *Beta* sect. *Corollinae*, including *B. lomatogona* Fisch. & C.A. Meyer, *B. macrorhiza* Stev., *B. corolliflora* Zosimovic ex Buttler, *B. trigyna* Waldst. & Kit., *B. nana* Boiss. & Heldr. [6,21,25]. Out of 11 wild species in the genus *Beta*, 5 have been recorded in the territory of the Republic of Armenia [26,27,28,29,30,31]. These species include *B. vulgaris* subsp. *maritima* (L.) Arcang (*B. perennis* (L.) Freyn.), *B. macrorhiza* Stev., *B. trigyna* Waldst. & Kit., *B. lomatogona* Fisch. & C.A. Meyer and *B. corolliflora* Zosimovic ex Buttler. They have been found in the floristic regions of Shirak, Aragats, Aparan, Lori, Tavush, Sevan, and Darelegis (Table 1). However, the information about *B. vulgaris* subsp. *maritima* and *B. trigyna* is very scarce. The most often revised species are *B. lomatogona*, *B. corolliflora* and *B. macrorhiza* (Figure 1A–D, Table 1) [26,27,28,29,30,31]. 

Despite the fact that some records for the wild beets in Armenia are found in the literature and databases, no general overview, morphological characterization and species distribution in recent years has been complied. In this paper, we summarize taxonomic background, morphological features, and geographic distribution of wild beet species of Armenia. Annual monitoring of the species in 2016–2021 in four different locations is presented (Figure 2). The research objectives discussed are:Distribution of the biodiversity of the wild beet species in the context of the impact of climate change over the last decade.Identification of *Beta x intermedium* Aloyan, the new natural hybrid between *B. corolliflora* and *B. macrorhiza*, and its phylogenic relation with the wild beet species recorded in the region.Conservation measures for the preservation and recovery of endangered or have been lost and may have become extinct genetic resources.

## 2. Results

### 2.1. Distribution and Conservation

Over the last two decades, numerous fieldworks in *Beta* habitats were conducted by our group for observing the growth and distribution of the wild beets. Overall, three species were identified: *B. lomatogona*, *B. corolliflora* and *B. macrorhiza* (Figure 1, Table 1), and the selected locations for each species were observed annually for the last 5 years (Figure 2). *B. lomatogona* was monitored in its natural habitats in the floristic region of Shirak (Figure 1B). During the annual fieldworks in 2014–2019, no *B. lomatogona* plants were found around the villages of Akunq and Zarindja. The last fieldwork in July 2021, however, revealed one population of the species with 8 plants, located at (40°41′ N, 43°90′ E) ca. 100 m away from the previous collection site at 89 m higher elevation (Figure 1A, Table 2).

*B. corolliflora*, the most frequent wild beet species in Armenia, is found in the floristic regions of Sevan, Aparan, Lori, Ijevan, Yerevan, and Darelegis (Figure 1C). The annual observations around Hrazdan (40°39′ N, 44°66′ E) in the Kotayk region showed that the plants are preserved at the altitudes of 1675 m a.s.l. The number of plant populations has decreased in recent years, as they are preserved in narrow gorges, mostly on the northern slopes, and rocky locations. Along with the decrease in the number of populations in the area, the population size increased from 18, 22, 24 to 30 annually in 2016–2019 (Table 2). Our records of *B. corolliflora* in Tsakhkadzor (40°53′ N, 44°71′ E) and near Aghavnadzor (40°57′ N, 44°68′ E) were at 1825 and 1750 m a.s.l., respectively (Table 1). In 2021, the species was also located in the region of Mount Ara (40°40′ N, 44°46′ E) at 2360 m a.s.l. (Table 1). Despite the record of *B. corolliflora* in the same region from 1980 (Table 1), our group was unable to find the species in the area during the previous years.

*B. macrorhiza* was found at the altitudes of 1400–2200 m a.s.l. mostly in the Darelegis and Sevan floristic regions in Vardenyats (Selim) Mountain Pass, near Gladzor (Vayots Dzor marz), around the Gyumri-Spitak highway and in the region of Sevan Lake (Table 1, Figure 1D). The species distribution has declined over the years, however. We were not able to find plants in some locations around Lake Sevan, where the species were reported from the 1980s to early 2000 (Table 1). The fieldworks revealed that the species registered in the Vardenyats Mountain Pass during earlier studies were still present (Table 1). The number of plants within the population was gradually increasing from 2016 to 2019 (Table 2). Species populations were found also at the altitude of 2410 m a.s.l. (39°93′ N, 45°23′ E), which is 200–300 m higher than the earlier records (Table 1, Figure 1D).

All the three species were collected as herbarium specimens and registered in the European Search Catalogue for Plant Genetic Resources (EURISCO) international database: *B. lomatogona* 8L01–8L05 (5 accessions), *B. macrorhiza* 8M01-8M05 (5 accessions), and *B. corolliflora* 8C01-8C19 (19 accessions). The seeds were placed for long-term ex situ conservation at −18 °C in the National Genbank of Crops and Crops’ Wild Relatives of Agrobiotechnology Scientific Center, ANAU Foundation.

### 2.2. Morphology 

The three species *B. lomatogona*, *B. corolliflora* and *B. macrorhiza*, collected, identified and registered during our fieldworks, are perennial herbaceous plants (Figure 3, Figure 4 and Figure 5). 

All of them refer to hemicryptophytes with annual monocyclic monocarpic shoots. They form a leaf rosette in the first year and bolt the following years, developing generative shoots with inflorescences and petiolate lower and sessile or sub-sessile upper leaves. Flowering and fruiting are observed in July–August, after which the shoots dry. The storage root differs in shape and size between the species. The morphological features of the species are presented in Table 3.

### 2.3. Identification of Unknown Beta Sample

During the fieldwork to the Vardenyats Mountain Pass in July 2019, a single *Beta* plant was spotted, located ca. 50 m away from the area populated with *B. macrorhiza* species, that had morphological differences from the *B. macrorhiza* earlier described in that region. We will refer to the plant as Unknown *Beta* in the text from this part on. The plant was shorter, approximately half the height of the neighbor *B. macrorhiza* plants, had a half-erect rosette, with the leaves spread out on the ground. Leaves were notably narrower with long petioles, darker in color, with a smooth and shiny surface. In addition, it had a reddish root (Figure 6). 

All recorded differences between the found unknown species and the neighbor *B. macrorhiza* plants are listed in Table 4.

Unknown *Beta* did not fully fit the identification features of any of the described wild beet species (Table 3), providing support for it being a new species, or at least genotype. To find out whether the observed differences were a result of high polymorphism within *B. macrorhiza,* or if it was a possible cross between the species, we used nuclear and chloroplast molecular markers to assess the samples’ genetic relationships through phylogenetic analyses. We also included the available *B. trigyna* and *B. vulgaris* subsp. *maritima* sequences from the Genbank: the latter was earlier recorded in the area within the same floristic region, though at a lower altitude, and *B. trigyna* was recorded around the same locality at a similar elevation.

When comparing the phylogenetic relation between the collected samples, the Unknown *Beta* was most similar to *B. macrorhiza* (BP = 96) by nuclear sequence, while by chloroplast sequence it showed the closest similarity to *B. corolliflora* (BP = 98). The addition of sequences from the GenBank to the tree showed that the Unknown *Beta* phylogenetically stands between *B. corolliflora* and *B. macrorhiza,* based on the chloroplast marker sequence similarity (Figure 7D). We could suggest that the collected plant is a hybrid between these two species that share the same growth area (Table 1, Figure 1C,D). Morphological features that resemble *B. corolliflora* include long petioles and leaf shape (Table 3). The other characteristics, including the reddish root, could be specific features derived from a genetic and/or environmental effect. Based on the morphological and genetic analysis we suggest a possibility for the Unknown *Beta* to be considered as a new species *Beta x intermedium* Aloyan. Yet, additional evaluation is needed for the correct identification of the rank of the new taxon. 

The phylogenetic analysis showed that the local *B. corolliflora* was matching the *B. corolliflora* from the database, though the percentage of similarity was low (BP = 28). An interesting result came with *B. macrorhiza* being in closer relation to *B. trigyna*, rather than to *B. macrorhiza* from the GenBank.

We could not find the sequences of the *adh* gene for *B. corolliflora* and *B. trigyna* in the GenBank, hence, the phylogenetic analysis for the nuclear sequence was performed without involving these two species. The unknown *Beta* was more similar to *B. macrorhiza* using the nuclear DNA, and interestingly closer to the *B. macrorhiza* from the GenBank than to the local sample (Figure 7C). 

*B. vulgaris* subsp. *maritima* formed a distinct monophyletic lineage from the other species of the *Corollinae* section by chloroplast sequence, however, the *adh* gene seems to be more conserved in all the species. Nevertheless, the closest relation to the *B. vulgaris* subsp. *maritima* showed *B. lomatogona* in both the nuclear and chloroplast loci. The *B. lomatogona* among the analyzed species was the closest to the *B. lomatogona* from the GenBank in both nuclear and chloroplast DNA (BP = 98), thus, showing a divergence from the other species. 

## 3. Discussion

Fieldwork conducted to observe the growth and distribution of the wild beet species revealed a general reduction in the populations in the native habitats. The most severe reduction was recorded for *B. lomatogona.* The species grows in dry semi-deserts climate in steppes, which are most prone to the impacts of climate change. A continuous decrease in the population sizes was reported by our group earlier [34]. The findings presented here show no recorded plants around two of the limited native habitats of the species in Akunq and Zarindja villages in 2014–2019. Aragatsotn marz, where the villages are located, is the most arid natural habitat of the local wild beets (Figure 8A,C). The average annual precipitation was highly variable between 2010 and 2020, with the 2013 rainfall less than half that reported in 2010 but increased to previous figures by 2016 before again declining to the 2013 rainfall level by 2020 (Figure 9A). Along with temperature changes, these climatic factors likely affected plant growth. Interestingly though, a small population of *B. lomatogona* was found in 2021 (Table 2). This could be connected to the relative high precipitation and warmer temperatures in the winters of 2018–2020 (Figure 8A,B, Appendix A), which caused a favorable environment for seed germination and plant growth [35,36]. Notably, the new population was located at a higher altitude from the originally recorded location. A vertical shift of the limits of the flora distribution up the mountain profile due to climate change was reported in Armenia [2]. The changes in population size and geographic location of *B. lomatogona* could be also affected by grazing [37,38]. The area of the species distribution was commonly used by the livestock from the neighboring villages. Nevertheless, the climate factors are likely to be the most important factors affecting the observed effects. *B. lomatogona* was included in the *Red Book of Plants of*
*the Republic of Armenia* in 2012 [33], and is currently considered as a critically endangered species, which therefore needs particular measures for preservation. 

Changes in the distribution of *B. corolliflora* in terms of alterations of the growth area and density of the population were observed, though generally being less drastic compared to *B. lomatogona.* Originally being predominant in humid areas at the altitudes of 2000–2700 [27], *B. corolliflora* was also found at 1400–2360 m a.s.l. [28,39]. It should be mentioned that records of the species at the low altitudes by roads may not be considered as species’ natural habitats, as they were likely dispersed to the lower areas by humans. We showed here that in the selected location the species was preserved at the same altitude during 2016–2021. However, as a general observation, *B. corolliflora* has been found recently at ca. 100–150 m lower altitudes than in the 1980s (Table 1). 

Precipitation and temperature irregularities have more pronounced effects in arid and cold biomes than in wet and temperate ones [36]. The climate in Kotayk is cooler and more humid compared to Aragatsotn (Figure 2). The gradual temperature increase of around 1.7 °C observed since 2011 (Figure 9B) in the Kotayk region possibly sustained plant growth and development and might also have supported the migration of the species from upper to middle mountain belts. The reduction of the average annual precipitation in 2017–2020 (Figure 9B) could have been detrimental for the species [36]. However, the high precipitation level in December–March of 2017–2019 and slightly cooler summers in 2019–2020 (Figure 8A,D, Appendix A) have probably mitigated the adverse impact of the increasing drought [40,41]. We also suggest that the changes in the distribution of *B. corolliflora* during the past several years were not associated only with the changing climate. Hrazdan region is a heavily exploited zone with intensive agriculture and increased urbanization, so the involvement of the anthropogenic factors should be seriously considered [42,43].

The third species, *B. macrorhiza*, has isolated growth areas (Figure 1D). The extent of occurrence and the areas of occupancy are less than 500 km^2^. *B. macrorhiza* is included in the Red Book of Plants of the Republic of Armenia as a vulnerable species [33]. The species was reported to grow on middle and upper mountain belts at the altitudes of 1400–2200 m a.s.l. on dry stony slopes, in steppes and meadow-steppes, and on sandy banks of the lakes [27,28,33,39]. Vardenyats Mountain Pass in Gegharkunik, the selected location for the annual growth observation of *B. macrorhiza*, is the highest geographic locality for the wild beets in Armenia. Stretching of the natural habitat to the altitude of 2410 m, recorded in the present study, could be related to climate change. The region is relatively colder, with lower precipitation in winter and more wet summers, compared to other selected locations (Figure 2). The average annual temperature fluctuation in 2011–2020 was 2 °C (Figure 9C). The average annual precipitation has gradually decreased since 2010, then increased in 2016 and 2018 (Figure 9C). With the exception of 2017, the precipitation in July-September 2016–2020 was higher than in the earlier years (Figure 8C, Appendix A), which could be related to the species dispersal to the higher altitude. 

The increase in population size observed in *B. corolliflora* and *B. macrorhiza* in 2016–2019 (Table 2) could be an adaptive response to the climate change for coping with genetic drift and inbreeding depression by increasing the possibilities for genetic variation within populations under the stress conditions [44]. High levels of heterozygosity have been repeatedly shown to confer resistance to environmental change, and intensify selection pressure for adaptation to climate change [45]. 

The wild beet species showed a high level of morphological polymorphism (Table 3), which is consistent with the other reports [27,39]. The differences in the sizes and the shapes of stems, leaves and roots, could be the morphological responses to the climate variation occurring [46,47]. Nevertheless, we cannot disregard the effect of the non-climatic factors underlying the morphological diversity within the species, including mutations, gene flow, genetic drift and/or differential gene expression and epigenetics, as a response to natural selection pressure [48,49]. 

The native habitats of the studied *Beta* species vary in geographic location, landscape, altitude and climate and comprise different ecosystems (Figure 2 and Appendix A). The mountain steppe and meadow-steppe ecosystems, where the species grow, are described as highly diverse in plant communities and rich in species composition, also involving crops’ wild relatives, endemic and rare species [1,2]. The projected changes in climate, such as increase in air temperature and reduced precipitation, will increase evaporation rates and reduce winter snowpack and spring run-off: as a result, less water will reach streams and rivers, leading, eventually, to the reduction in groundwater reserves, decline in water and soil quality and desertification [13,50,51]. Such changes will increase the expansion of desert, semi-desert, arid sparse forest areas and steppes at the expense of the vertical shift of their upper limits and reduce ecosystem productivity, resulting in species range shifts, and potential loss of biodiversity [14,15,52,53]. A reduction in the lower part of steppe belt has been already observed in Armenia due to the expansion of semidesert vegetation, along with penetration of typical steppe species into a meadow-steppe zone with a reduction in its altitudinal limits [1]. The expansion of invasive and expansive plant species, as a result of the climate change in addition to the anthropogenic impact, has been reported in the National Reports to the Convention on Biological Diversity [1,2]. Active soil cultivation and improper agricultural practices in steppes have an additional negative impact on the ecosystem’s biodiversity [1,2]. For the proper understanding of the impact of climate change on ecosystems, more data and a complex approach are needed in order to draw a broad comprehensive conclusion for short- and long-term effects.

We identified the Unknown *Beta* sample as a natural hybrid between *B. corolliflora* and *B. macrorhiza.* To our knowledge, this is the first documented hybrid described between *B. corolliflora* and *B. macrorhiza*. The close relation of *B. corolliflora* and *B. macrorhiza* by chloroplast marker sequence similarity (Figure 7B,D) may have facilitated the cross between the species. The close phylogenetic relation of these two species could be explained by partly sharing a distribution area within the Darelegis floristic region (Figure 1C,D). *B. trigyna* also shares the same area (Table 1), which explains its close relation to *B. macrorhiza* (Figure 7D). Further studies with a larger sampling and using more molecular markers would contribute to a deeper assessment of phylogenetic relations and gene flow between the species. 

Few natural hybrids between the wild beets have been reported. McFarlane [54] reported the *B. vulgaris* subsp. *maritima* × *B. macrocarpa* hybrids. Tetraploid *B. macrocarpa* was mentioned as a natural amphidiploid between diploid *B. macrocarpa* and an unknown diploid of the *B. vulgaris* complex [55]. *B. maritima* was reported to form natural hybrids with cultivated beet [56]. The majority of the beet hybrids, however, were produced between *B. vulgaris* and different wild relatives [6,57,58]. None of the reports on hybrids between *B. vulgaris* and the *Corollinae* section species made it clear whether an introgression through natural recombination had occurred [58]. Pivovarov and Burenin, in addition, described the crosses between wild species being difficult [57]. There are several reports discussing the effect of climate change on interspecies hybridization [59,60]. Climate warming in the high-altitude areas would facilitate species migration in acquiring new habitats and increase outcrossing between the species. However, so far, the information on the effect of climate change on the interspecies hybridization in the genus *Beta* is scarce. 

Natural hybridization is an important process in plant evolution that provides gene flow between genetically different populations or taxa, and may influence the adaptive response to selection, leading to the formation of a new species [61]. The role of gene expression for speciation is still relatively unexplored, despite the increasing number of studies characterizing candidate genes or describing general patterns of gene expression related to adaptive divergence in different systems [62]. 

Landscape features, both biotic, abiotic, natural and/or anthropogenic, may interfere with migration and gene flow, affecting the level of genetic differentiation among populations [63] (and ref. therein). Considering the complexity of the landscape in Armenia, geographical isolation is likely to be a factor in *Beta* species differentiation. *B. lomatogona* is the most geographically distant and the most genetically divergent species compared to the other local wild *Beta* species (Figure 1 and Figure 7). *B. macrorhiza* has isolated growth areas in Darelegis, Lori, Shirak and Sevan, generally being isolated from the *B. corolliflora* populations distributed in Aparan and in the north part of Sevan floristic regions (Figure 1). The overlapping growth area of *B. macrorhiza* and *B. corolliflora,* where the *Beta x intermedium* Aloyan was found, has a difficult topography, where species are located on the opposed mountainous slopes of different elevation that are exposed to different climate conditions. Therefore, ecological and geographical barriers help to keep these taxa mostly separated [64,65,66,67]. Species with smaller ranges, and those with narrow and fragmented habitats are more vulnerable than others. In this sense, *B. lomatogona* and *B. macrorhiza* face, at a different degree, severe survival problems, being, in addition, exposed to the continuous climate change impact and to the threats of human activities, such as, for instance, uncontrolled animal grazing [2,68]. 

Wild and weedy forms of *Beta vulgaris* subsp. *maritima* (Table 1) were reported in Armenia on lowlands and foothills on slightly saline soils in the south-east part of Yerevan [26], in the Ararat valley and on red clays or on stony slopes in the Ijevan and Darelegis floristic regions, at altitudes ranging from 800 to 1800 m [28,31]. Despite being mentioned in the literature, the species was not conserved or registered in the catalogs, and no recent findings of the species were recorded. *B. vulgaris* subsp. *maritima* is usually distributed along the sea shores [69,70,71]. Inland populations are common for the Mediterranean basin, where they grow in deserted areas, clay soils and soil with high salinity [72,73,74]. The areas that the species were recorded in Armenia have climatic features rather different from the main distribution area of the sea beet in the Mediterranean area or in Western Europe. It is yet unclear how the sea beet survives under these conditions in Armenia. Phylogenetic divergence of *B. vulgaris* subsp. *maritima* from other *Beta* species based on the chloroplast DNA marker could be explained by the taxonomic distance of the species, belonging to the different *Beta* section (Figure 7B,D). The alcohol dehydrogenase sequence seems to be well conserved in all the analyzed *Beta* species (Figure 7A,C).

The *B. trigyna* species was first registered in 1928–1929 (Table 1). The species was later included in *B. corolliflora* by Buttler [75] and Takhtajian [26]. Molecular analysis, however, does not support this inclusion as the species appears to be more closely related to *B. macrorhiza* than to *B. corolliflora* [25,76]. The same conclusion was drawn from our studies. The *B. trigyna* from the database was closer to our collected *B. macrorhiza* rather than to *B. corolliflora* (Figure 7D). Detailed molecular analysis would help to reveal the distribution of *B. trigyna* in Armenia. 

## 4. Materials and Methods

### 4.1. Study Design and Sample Collection

Four locations of natural habitats of *Beta corolliflora*, *Beta macrorhiza* and *Beta lomatogona* (Figure 2) were randomly selected for the annual growth monitoring during the species flowering–fruiting period (July–August) in 2016–2019 and 2021 (2020 was dropped due to the COVID-19 pandemic). The sites included the neighborhoods of Akunq and Zarindja villages in Aragatsotn marz (Shirak floristic region) for *Beta lomatogona* (Plots 1 and 2, respectively), localities in Hrazdan in Kotayk marz (Aparan floristic region) for *Beta corolliflora* (Plot 3) and areas around Vardenyats Mountain Pass in Gegharkunik marz (Sevan floristic region) for *Beta macrorhiza* (Plot 4) (Figure 2). The selected areas represent different geographic and climatic zones that have been subjected to the climate change impact to various degrees [8,77]. Two localities for *B. lomatogona* were chosen due to the limited population distribution of this species. In these two locations, the fieldworks were organized also in 2014 and 2015. The observation sites were ca. 1000 m^2^. The study of populations was carried out on the principle of complete observation of the selected area; the population size was determined by counting the individuals in the plot. Species samples from these locations were collected for morphological evaluation and conservation as herbarium specimens. Fresh leaves were collected and kept at −20 °C for DNA extraction. The species collection sites were mapped with GPS. The plant material was preserved by drying on a flow of hot air at a maximum temperature of 50 °C, and deposited at the National Gene Bank Herbarium Collection, ANAU, Yerevan, Armenia. In 2017–2018 the fieldwork was also organized in September–October for seed collection and conservation purposes. The obtained seeds were stored at −18 °C. The collected information about the distribution of the wild beets in the selected locations was aligned with the temperature and precipitation data obtained from the Hydrometeorology and Monitoring Center, State Non-Commercial Organization (SNCO) of the Ministry of Environment of the Republic of Armenia. 

In addition to the mentioned locations, seasonal fieldwork by our group in different natural habitats of the wild beets has continued since 2001, collecting species distribution data and developing reliable maps of occurrence of the various *Beta* species in Armenia. The findings from this fieldwork are included in Table 1. The morphological features of the species from these observations are summarized in Table 3.

### 4.2. Morphological Study

Morphological evaluation of the samples included height of the plant (cm), shape of stems, shape of rosette, shape of leaves, leaf length and width (cm), petiole length (cm), root shape, length/diameter (cm) and color, flower structure, and weight of glomeruli. Student’s *t*-test (*p* ≤ 0.05) was used for analyzing morphological differences of the samples using Microsoft Excel 2010.

### 4.3. DNA Extraction, Amplification and Sequencing

Prior to DNA extraction, dry or fresh plant material was ground to a fine powder with liquid nitrogen, using a mortar and pestle. DNA was extracted from 100 mg of plant powder using the E.Z.N.A.^®^ HP Plant DNA Kit (Omega Bio-tek, Norcross, GA, USA), following the manufacturer’s instructions. The concentration and purity of DNA samples were measured using NanoDrop 2000 spectrophotometer (Thermo Scientific, Foster City, CA, USA). The integrity of DNA was assessed using 1% agarose gel electrophoresis. The extracted DNA was stored at −20 °C until further use.

Two regions largely used in phylogenetic studies were selected for sequencing. These barcodes were previously successfully used by Touzet et al. on the same family [70]. The chloroplast DNA (cpDNA) trnL-trnF intergenic spacer (LF) with primers (forward/reverse) 5′-GGTTCAAGTCCCTCTATCCC-3′/5′-ATTTGAACTGGTGACACGAG-3′ (annealing temperature [T_a_] = 55 °C) [78] and the partially sequenced alcohol dehydrogenase (*adh*) of nuclear DNA with primers 5′-TGTCCTGCCTGTTTTCACTG-3′/5′-TACTGCTCCTAGGCCGAAAA-3′ (T_a_ = 55 °C) anchored in exons 1 and 2 [70] were used. PCR amplification was performed in 25 μL reactions, containing 200 ng of DNA template, 2.5 μL of DreamTaq Buffer 10X (Thermo Fisher Scientific, Baltics, Vilnus, Lithuania) 1 μL of each primer (5 μM), 0.5 μL of dNTP (10 mM) and 0.5 μL of DreamTaq DNA polymerase (Thermo Scientific, Foster City, USA). The PCR was performed using an MJ MiniOpticon instrument (BioRad, Hercules, CA, USA) according to the following program: an initial denaturation step at 98 °C for 30 s, followed by 35 cycles of denaturation at 98 °C for 10 s, annealing at 55 °C for 15 s, extension at 72 °C for 30 s, a final extension at 72 °C for 5 min, and a holding step at 4 °C. 

The PCR products were visualized using gel electrophoresis; the target bands were sliced out of the gel, purified using an E.Z.N.A. Gel extraction Kit (Omega Bio-tek, Norcross, GA, USA) and sequenced using BigDye Terminator v3.1 Cycle Sequencing Kit (Applied Biosystems, Foster City, CA, USA). Amplification was performed in a 20 μL reaction mix containing 2–4 ng of DNA template, 1 μL of Big Dye (RRM), 3.5 μL of 5X Sequencing Buffer, and 1 μL of primer (1 μM) using a MJ MiniOpticon thermal cycler (BioRad) with 29 cycles at 96 °C for 30 s, 55 °C for 15 s, and 60 °C for 4 min. 

### 4.4. Sequence Alignment and Phylogenetic Analysis

Taxonomic identification was assigned based on the best BLAST hit to a sequence with ≥97% identity, e-value ≥ 10^−5^, and minimum query coverage > 84–91%. Raw data of sequences were read, verified, and aligned using the Clustal Omega multiple sequence alignment software [79]. The analyzed sequences were deposited at the National Center for Biotechnology Information (NCBI) GenBank database. All the sequences used in generating the phylogenetic trees are listed in Table 5. The aligned sequences were used for the construction of phylogenetic trees using the maximum likelihood (ML) algorithm implemented in Mega X Molecular Evolutionary Genetics Analysis across computing platforms [80].

## 5. Conclusions and Recommendations

Climate change affecting Armenia varies substantially from region to region and from season to season. Hence, the changes in biodiversity expected under the changing climate conditions vary from region to region. Despite the average annual increase in temperature and general decrease in average annual precipitation over recent years, there are some fluctuations in temperature and precipitation levels from year to year, along with seasonal variations, which help plants to adapt to changing climate conditions. Migration, a high degree of polymorphism, and the occurrence of outcrosses between the species show a potential for genetic diversity as a tool for survival. We showed that the wild beets *B. lomatogona*, *B. corolliflora* and *B. macrorhiza* are sensitive to climate changes and were affected to various degrees, depending on their location. The most affected species was *B. lomatogona*, which is at the verge of extinction. Migration up the mountain belt was recorded for *B. lomatogona* and *B. macrorhiza*, and *B. corolliflora* was found at lower altitudes than in the 1980s. A general reduction in the beet’s population size in the native habitats was observed, with an increased number of plants within the populations, recorded for *B. corolliflora* and *B. macrorhiza*. A new natural hybrid *Beta* x *intermedium* Aloyan between *B. corolliflora* and *B. macrorhiza* was described. The available data from this work was not sufficient for precise evaluation of the effect of the climate change on the ecosystem level.

A combination of broader ecological and genetic data is providing a solid foundation for better understanding of the phylogeny and distribution of Armenia’s wild beets as a basis for conserving this important element of the country’s natural heritage and helping this biodiversity to contribute to the changing environmental conditions being driven by accelerating climate change.

In the face of the climate change impact, continuous monitoring of species distributions is necessary for determining the state of the habitats in danger of destruction and assessment of vulnerability of ecosystems in biodiversity-rich areas [2]. Among the other national targets used for following up the Convention on Biological Diversity is the enhancement of in situ and ex situ conservation of the biological diversity [2]. The network of specially protected nature areas in Armenia is continuously growing thanks to the state support [2]. Nevertheless, improvement of the conservation management, expansion and upgrade of gene bank equipment along with enlargement and enrichment of the genetic resources collection are necessary [2]. Legislation improvement, adoption of special national programs on crop wild relatives conservation may support species preservation and recovery of the endangered or destructed genetic resources [2,82,83,84]. Due to the adoption of a number of legal acts within recent years, the key issues and directions of management of natural resources are set, the risk of non-sustainable use of resources at the initial stage of economic activities could be avoided, along with prevention of respective forms of activities (mine exploitation, etc.) in certain areas, where valuable biodiversity representatives, landscapes, and species registered in the Red Books of RA are present [2]. Measures used to prevent the uncontrolled use of bioresources via new effective forms of inter-sectoral cooperation for better follow-up of the regulations related to the protected areas would be appropriate. State support of education of the population for understanding the importance of biodiversity preservation could work as a part of a national program for local species preservation. 

Despite the tackled ex situ conservation of the wild *Beta* species of Armenia, their in-situ conservation has not been achieved yet. The protection of the ecosystems where wild *Beta* is found in Armenia should be organized, especially for *B. lomatogona* and *B. macrorhiza* species, that are registered in the *Red*
*Book of Plants of the Republic of Armenia*. It can be best implemented if these ecosystems are introduced into the list of specially protected nature areas by the government decision under the “natural monuments” category, complying with the international standards.

## Figures and Tables

**Figure 1 plants-11-02502-f001:**
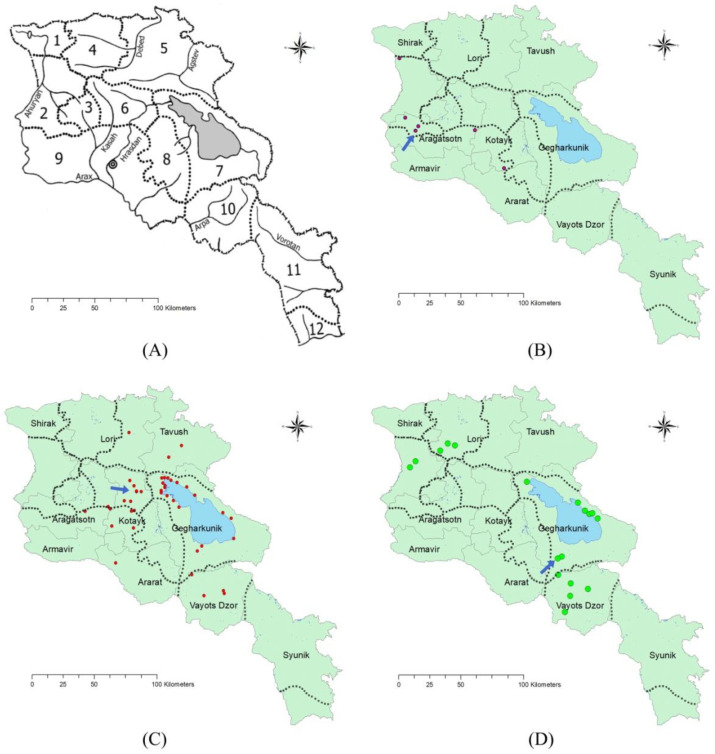
Maps of the floristic regions of Armenia (**A**): 1—Upper Akhuryan; 2—Shirak; 3—Aragats; 4—Lori; 5—Ijevan; 6—Aparan; 7—Sevan; 8—Gegham; 9—Yerevan; 10—Darelegis; 11—Zangezur; 12—Meghri; and geographical locations of the three *Beta* wild species in Armenia: *B. lomatogona* (**B**), *B. corolliflora* (**C**) and *B. macrorhiza* (**D**). The selected locations for annual observations described in this paper are shown with blue arrows. The distribution range of the species is shown in Appendix A [32].

**Figure 2 plants-11-02502-f002:**
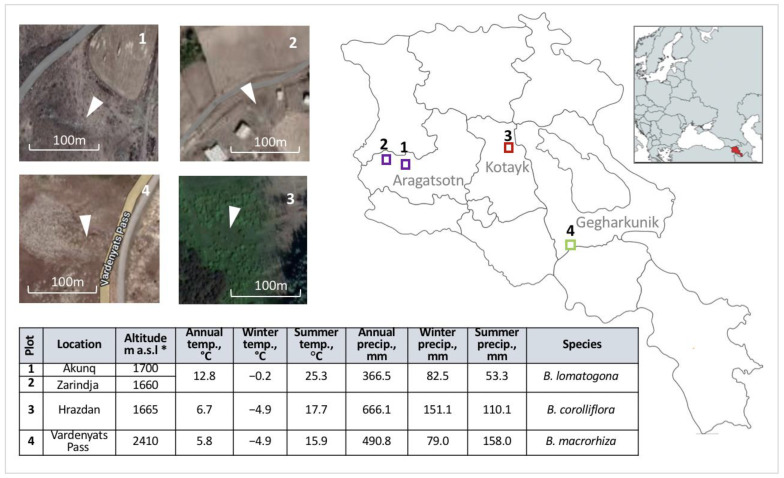
Study area and sampling localities (Plots 1, 2, 3 and 4) for annual monitoring of the wild beet species in 2016–2021 and climate parameters of the selected regions for the period of 2010–2020. * meters above sea level.

**Figure 3 plants-11-02502-f003:**
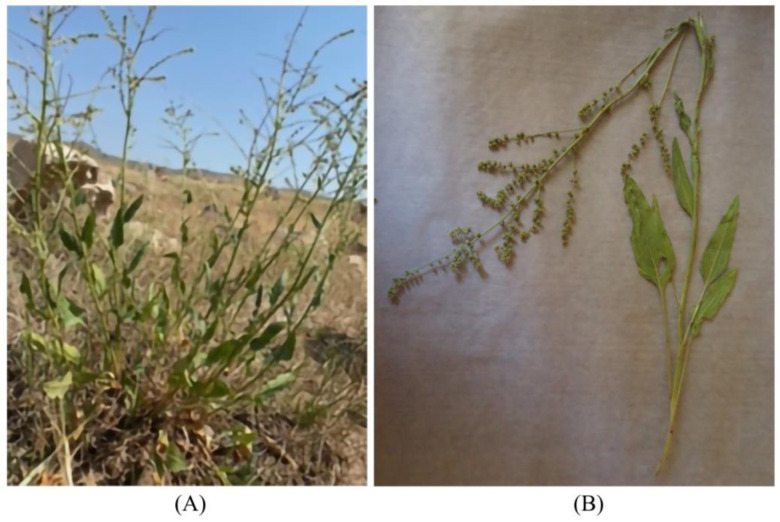
*Beta lomatogona* species in their natural habitat (**A**) and a herbarium specimen (**B**). Akunq village (1789 m a.s.l.), Aragatsotn marz, the Republic of Armenia (RA), 2 July 2021.

**Figure 4 plants-11-02502-f004:**
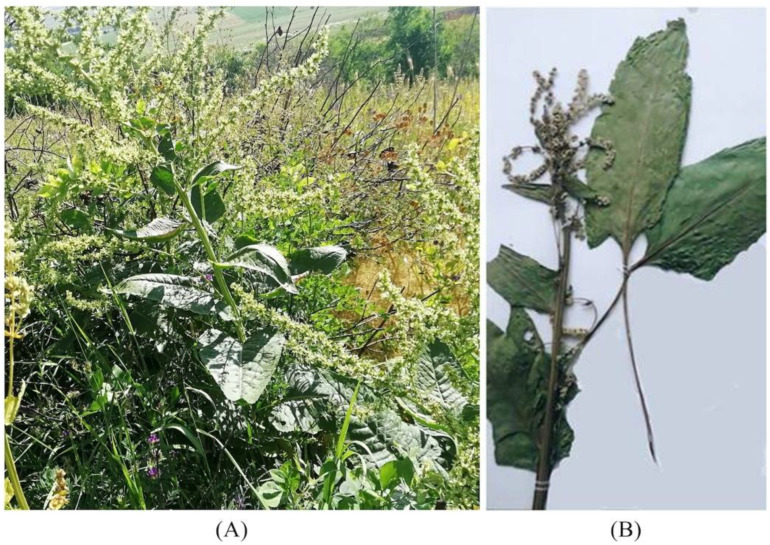
*Beta corolliflora* species in their natural habitat (**A**) and a herbarium specimen (**B**). Hrazdan town (1675 m a.s.l.), Kotayk marz, RA, 20 July 2019.

**Figure 5 plants-11-02502-f005:**
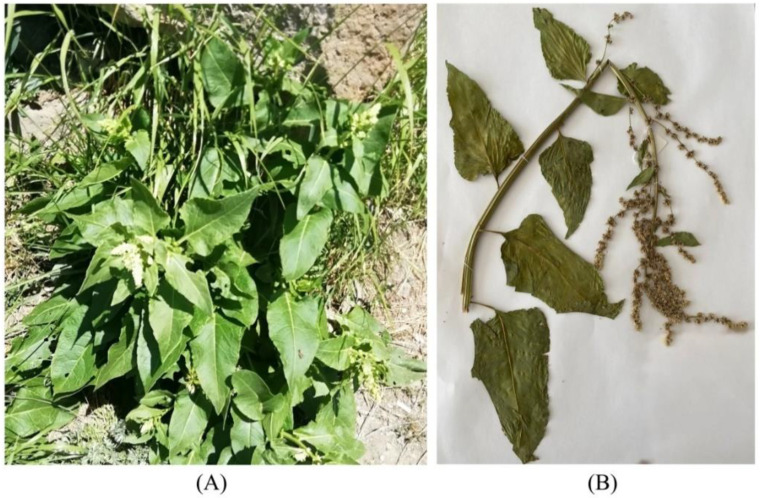
*Beta macrorhiza* species in their natural habitat (**A**) and a herbarium specimen (**B**). Vardenyats Mountain Pass (2410 m a.s.l.), Gegharkunik marz, RA, 12 July 2019.

**Figure 6 plants-11-02502-f006:**
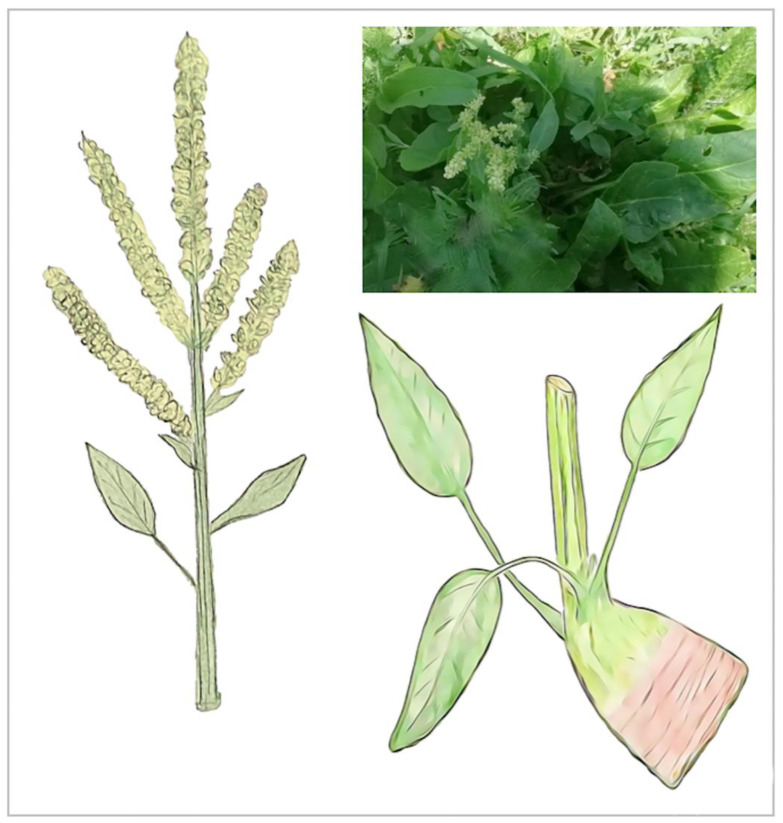
Unknown *Beta* sample, Vardenyats Mountain Pass, Gegharkunik marz, RA.

**Figure 7 plants-11-02502-f007:**
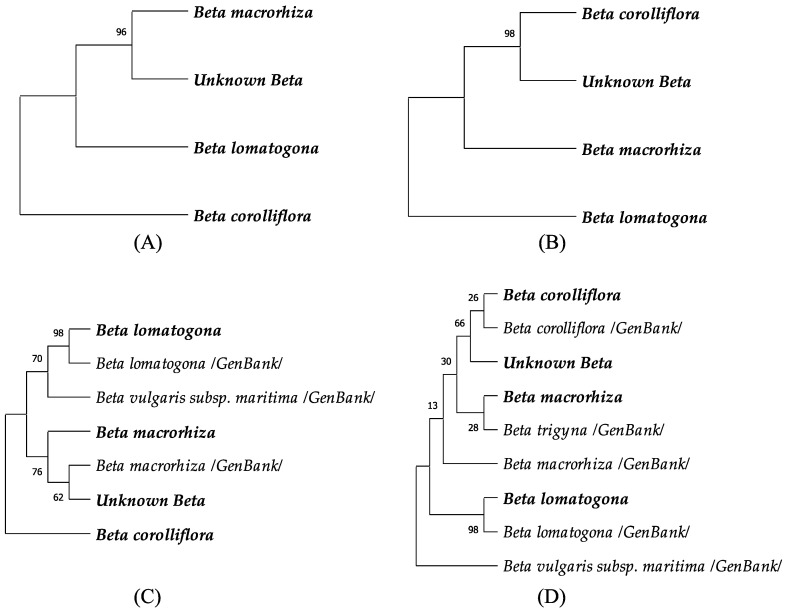
Phylogenetic relation of the wild beet species using maximum-likelihood (Tamura-Nei model) analysis, based on the partially sequenced alcohol dehydrogenase (*adh*) nuclear (**A**,**C**) and the trnL-trnF intergenic spacer (LF) chloroplast (**B**,**D**) markers. The numbers next to the branches indicate bootstrap percentages of 1000 replications (BP).

**Figure 8 plants-11-02502-f008:**
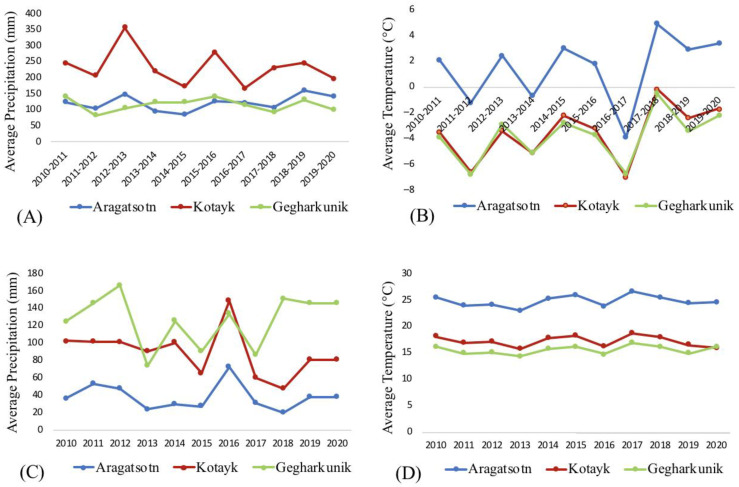
Total precipitation and average temperature during winter (**A**,**B**) and summer (**C**,**D**) periods in the selected regions of Aragatsotn, Gegharkunik and Kotayk in 2010–2020.

**Figure 9 plants-11-02502-f009:**
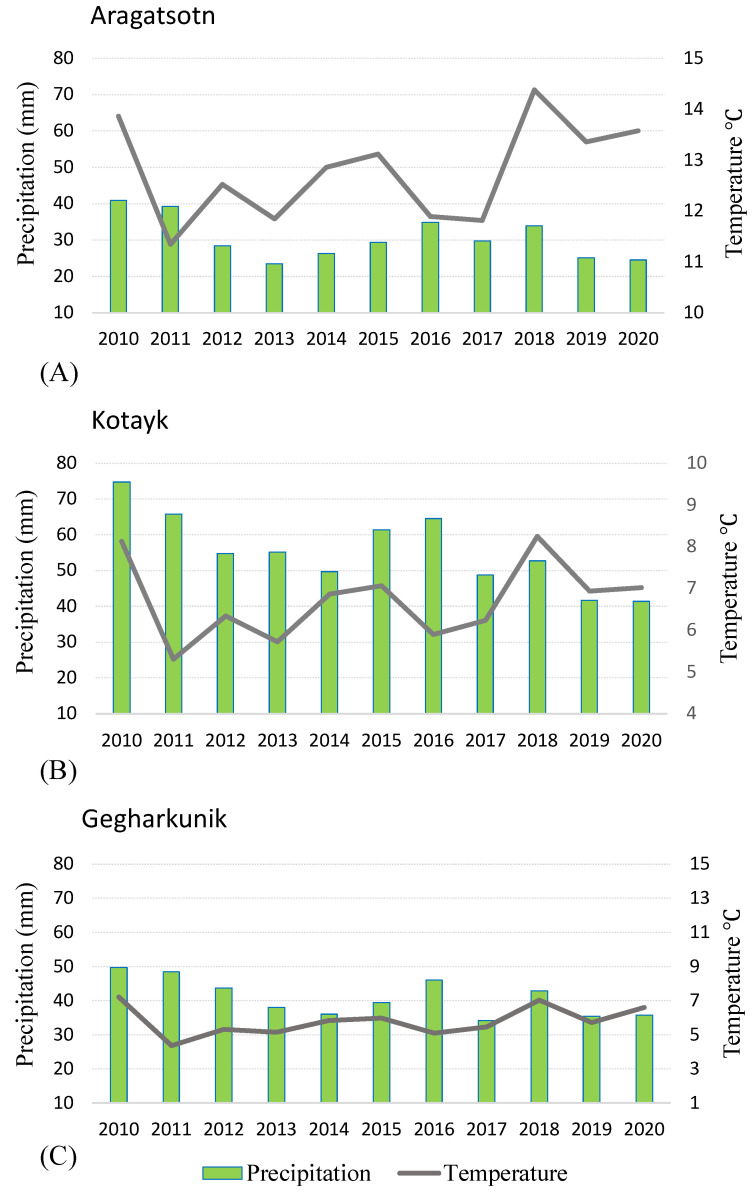
Average annual temperature and precipitation in Aragatsotn (**A**), Kotayk (**B**) and Gegharkunik (**C**) marzes of Armenia in 2010–2020.

**Table 1 plants-11-02502-t001:** Wild species of beet reported in Armenia. The table summarizes records of the wild beet species in Armenia, including information from the fieldworks of our group, accessible databases, catalogs, and published articles.

Species	Floristic Region	Province (Marz)	Location	Collection Year	Altitude, m a.s.l.	Source and/or Reference
** *B. lomatogona* **	Shirak	Aragatsotn	Zarindja	1988	1660	EURISCO catalog, Acc. № 8L03, [27]
			Akunq	1987	1700	EURISCO catalog, Acc. № 8L01, [27]
				2006	1700	EURISCO catalog, Acc. № 8L04, *This study*
				2021	1789	EURISCO catalog, Acc. № 8L05, *This study*
		Shirak	Dzithankov	1987	1740	EURISCO catalog, Acc. № 8L02, [27]
			Gtashen	-	1880	[28]
	Yerevan	Yerevan Kotayk	North from Yerevan	1933	1500	[29]
				1956	1390	[26,28]
	Aparan	Kotayk	Mount Ara	2002	1990	[30]
** *B. corolliflora* **	Ijevan	Tavush	Haghartsin, Hovq	-	1300	[28]
	Lori	Lori	Dsegh	1987	1350	EURISCO catalog, Acc. № 8C06; [27]
	Darelegis	Vayots Dzor	Gladzor	1987	2000	EURISCO catalog, Acc. № 8C07
			Karmrashen	1980	2065	EURISCO catalog, Acc. № 8C01
				1999	-	[29]
			Between Karmrashen and Herher	2003	1820	[30]
	Aparan	Aragatsotn	Aparan reservoir	1988	1800	EURISCO catalog, Acc. № 8C11; [27,28,31]
			Amberd castle	1988	2300	EURISCO catalog, Acc. № 8C12; [27]
				2001		[29]
			Mount Ara	1980	2240	[30]
				2021	2360	*This study*
		Kotayk	Teghenik	1987	1600	EURISCO catalog, Acc. № 8C02; [27]
			Akunq	1988	1400	EURISCO catalog, Acc. № 8C13
			Fantan	1987	1800	EURISCO catalog, Acc. № 8C04; [27]
			Hatis	1987	-	EURISCO catalog, Acc. № 8C05; [27]
			Hrazdan	1956	-	[29]
				-	1550	[30]
				2016- 2019	1675	EURISCO catalog, Acc. № 8C18; *This study*
				2021		EURISCO catalog, Acc. № 8C19; *This study*
			Aghavnadzor	2017	1750	EURISCO catalog, Acc. № 8C16; *This study*
			Meghradzor	1987	1960	EURISCO catalog, Acc. № 8C03; [27]
			Tsakhkadzor	1984	1980	[30]
				2017	1825	EURISCO catalog, Acc. № 8C17; *This study*
			Arzakan	2002	1670	[30]
	Yerevan		Abovyan	2002	1630	[28,29]
	Sevan	Gegharkunk	Sevan Lake district	1989	1900	EURISCO catalog, Acc. № 8C14; [27]
				2007	-	[29]
			Pambak	1987	1985	EURISCO catalog, Acc. № 8C09; [27]
				2000	1985	EURISCO catalog, Acc. № 8C15; [27]
			Areguni mountain range	1987	2220	EURISCO catalog, Acc. № 8C08; [27]
			Vardenyats (Selim) Mountain Pass	1987	2100	[30]
			Tsovagyugh	1988	2023	EURISCO catalog, Acc. № 8C10; [27]
			Areguni	-	2000	[28]
			Vaghashen	-	1956	[28]
			Geghhovit	-	2080	[28]
** *B. macrorhiza* **	Sevan	Gegharkunk	Sevan (Areguni mountain range)	1956	2014	[26]
			Tsapatax	2001	1950	[30]
			Pambak	2000	1920	[30]
				2009	2180	[30]
			Vardenyats (Selim) Mountain Pass	1987	2200	EURISCO catalog, Acc. № 8M03; [27,31]
				-	2110	[28]
				2016- 2021	2200	*This study*
				2019	2410	EURISCO catalog, Acc. № 8M04; *This study*
				2021	2410	EURISCO catalog, Acc. № 8M05; *This study*
	Darelegis	Vayots Dzor	Gnishik	1956	2020	[26,33]
			Yehegis river gorge	1956	-	[26,33]
			Gladzor	1987	1350	EURISCO catalog, Acc. № 8M01; [27]
			road Goghtanik—Herher	2005	2070	[30]
	Shirak	Shirak	Gyumri-Spitak highway	1987	-	EURISCO catalog, Acc. № 8M02; [27]
	Lori	Lori				
** *B. trigyna* **	Aparan	Kotayk	Mount Ara	1928	2480	[30]
			Tsakhkadzor	1929	2265	[30]
		Aragatsotn	Amberd castle	-	2140	[30]
	Gegham	Ararat	Khosrov Forest State Reserve	-	2750	[30]
	Sevan	Gegharkunk	Sevan	-	2280	[30]
		Vayots Dzor	Vardenyats (Selim) Mountain Pass	-	2165	[30]
** *B. vulgaris* ** **subsp *maritima***	Ijevan		-	-	800–1800	[26,28,31]
	Yerevan		-	-		
	Darelegis		-	-		

**Table 2 plants-11-02502-t002:** Population size of the wild beet species in the selected locations in 2016–2021.

Species	Location	2016	2017	2018	2019	2020 *	2021
*B. lomatogona*	Akunq/Zarindja	0	0	0	0	-	8
*B. corolliflora*	Hrazdan region	18	22	24	30	-	27
*B. macrorhiza*	Vardenyats Mountain Pass	10	15	17	20	-	16

* no fieldworks were organized in 2020 due to the COVID-19 pandemic.

**Table 3 plants-11-02502-t003:** Morphological features of the wild beet species in Armenia.

		*B. lomatogona*	*B. corolliflora*	*B. macrorhiza*
**Stems**	Shape/ Cross-section	Erect stem, round with small ridges	Strong thick stem, angular	Erect or procumbent stem, angular
	Height (cm)	40–75	50–150	70–95
	Diameter (cm)	0.3–0.7	0.8–2.0	3.0–4.0
Rosette	Shape	With semi-erect leaves	With erect leaves	With erect or semi-erect leaves
	Leaves length (cm)	12–15	15–19	14–16
	Leaves width (cm)	2–5	7–9	5–7
	Petiole length (cm)	10–12	15–17	5–7
Leaves	Shape	Oblong, triangular, lanceolate, ovate; smooth and shiny surface	Oval, lanceolate (other types possible) with a cordate leaf base; weak fluffy surface	Spear-shaped, large, wide, with a blunt base; smooth surface
Roots	Shape	Narrowed, fusiform, woody	Cylindrical, woody	Cylindrical, conical, soft
	Length (cm)	75–100	100 (or more)	120–150
	Diameter (cm)	6–8	up to 15	8–15
	Color	Red brownish	Whitish	Whitish
Flowers/Seeds		Tepals are greenish, wide, white membranous, unequally dented at apex. Single flowers are on a long spiciform inflorescence and are arranged in tight axillary glomeruli. Weight of 1000 glomeruli 12–14 g.	Tepals are corolla-like whitish or yellowish, broadly open during flowering, slightly curved inward. Bracts are linear, not exceeding flowers. Inflorescences are compound, pyramidal, dichasial, with sympodial branching, and with branched leaf-shaped bracts. Flowers (2)3 coalescing into glomeruli. Weight of 1000 glomeruli 43–56 g.	Tepals are greenish, yellowish-greenish, flat, wide open. Bracts broadly ovate, exceeding the flowers. Inflorescences are elongated spikes. Flowers 4–6(8) coalescing into glomeruli. Weight of 1000 glomeruli 55–56 g.

**Table 4 plants-11-02502-t004:** Morphological differences between *B. macrorhiza* and the Unknown *Beta*.

Species	Stem Height (cm)	Rosette Shape	Leaves					Root	
			Shape	Surface	Length * (cm)	Width * (cm)	Petiole Length * (cm)	Diameter (cm)	Color
*B. macrorhiza*	90	Erect	Spear-shaped	Smooth	15 ± 1.1 ^a^	6 ± 0.7 ^a^	6 ± 0.7 ^a^	12	White
Unknown *Beta*	40	Semi-erect	Ovate	Smooth and shiny	14 ± 1.0 ^a^	4 ± 1.2 ^b^	9 ± 1.0 ^b^	8	Red

* The data on the leaf sizes include the mean value from all the leaves of the Unknown *Beta* plant and the means of the leaf parameters of the 6 closest, randomly chosen *B. macrorhiza* plants, ± standard deviation. Statistically significant differences (*p* < 0.05) are indicated by different superscript letters.

**Table 5 plants-11-02502-t005:** List of DNA fragments used for construction of phylogenetic trees.

Gene/ Fragment	Accession Number	Definition	Marked in the Phylogenetic Tree as	Description/Reference
*adh*	KP747951.1	*B. lomatogona* alcohol dehydrogenase (*adh*) gene, partial cds	*Beta lomatogona*/GenBank	[70]
*adh*	KP747950.1	*B. macrorhiza* alcohol dehydrogenase (*adh*) gene, partial cds	*Beta macrorhiza*/GenBank	[70]
*adh*	KP747982.1	*B. vulgaris* subsp. *maritima* isolate 33HAT alcohol dehydrogenase (*adh*) gene, partial cds	*Beta vulgaris* subsp. *maritima*/GenBank	[70]
*adh*	OM857605	*B. lomatogona* alcohol dehydrogenase (adh) gene, partial cds	*Beta lomatogona*	Location: Akunq, Aragatsotn *This study, collection of 2006*
*adh*	OM857606	*B. corolliflora* alcohol dehydrogenase (*adh*) gene, partial cds	*Beta corolliflora*	Location: Hrazdan, Kotayk *This study*
*adh*	OM857607	*B. macrorhiza* alcohol dehydrogenase (*adh*) gene, partial cds	*Beta macrorhiza*	Location: Vardenyats mountain Pass, Gegharkunik *This study*
*adh*	OM857608	*Beta x intermedium* Aloyan alcohol dehydrogenase (*adh*) gene, partial cds	Unknown *Beta*	Location: Vardenyats mountain Pass, Gegharkunik *This study*
LF	KP747769.1	*B. lomatogona* trnL-trnF intergenic spacer, partial sequence; chloroplast	*Beta lomatogona*/GenBank	[70]
LF	KP747770.1	*B.**macrorhiza* trnL-trnF intergenic spacer, partial sequence; chloroplast	*Beta macrorhiza*/GenBank	[70]
LF	AY858608.1	*B.**corolliflora* tRNA-Leu (trnL) gene, partial sequence; trnL-trnF intergenic spacer, complete sequence; and tRNA-Phe (trnF) gene, paritial sequence; chloroplast	*Beta corolliflora*/GenBank	[81]
LF	AY858605.1	*B. trigyna* tRNA-Leu (trnL) gene, partial sequence; trnL-trnF intergenic spacer, complete sequence; and tRNA-Phe (trnF) gene, paritial sequence; chloroplast	*Beta trigyna*/GenBank	[81]
LF	KP747794.1	*B. vulgaris* subsp. *maritima* isolate 33HAT trnL-trnF intergenic spacer, partial sequence; chloroplast	*Beta vulgaris* subsp. *maritima*/GenBank	[70]
LF	OM857609	*B. lomatogona* trnL-trnF intergenic spacer, partial sequence; chloroplast	*Beta lomatogona*	Location: Akunq, Aragatsotn *This study, collection of 2006*
LF	OM857610	*B. corolliflora* trnL-trnF intergenic spacer, partial sequence; chloroplast	*Beta macrorhiza*	Location: Hrazdan, Kotayk *This study*
LF	OM857611	*B. macrorhiza* trnL-trnF intergenic spacer, partial sequence; chloroplast	*Beta corolliflora*	Location: Vardenyats mountain Pass, Gegharkunik *This study*
LF	OM857612	*Beta x intermedium* Aloyan trnL-trnF intergenic spacer, partial sequence; chloroplast	Unknown *Beta*	Location: Vardenyats mountain Pass, Gegharkunik *This study*

## Data Availability

The DNA sequences of the studied wild beets are deposited to GenBank under the following accession numbers: OM857605, OM857606, OM857607, OM857608, OM857609, OM857610, OM857611, OM857612.

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
