# Peer review of "Distribution of Biodiversity of Wild Beet Species (Genus Beta L.) in Armenia under Ongoing Climate Change Conditions"

_plants, 2022, doi:10.3390/plants11192502_

Round 1

Reviewer 1 Report (New Reviewer)

Rev-plants-1873196

The article entitled ‘Distribution of Biodiversity of wild beet species (genus Beta 2 L.) in Armenia under ongoing climate change conditions’ is interesting and may be published in Plants with major corrections. The article deals with important issues related to the protection of centers origin of Beta genus diversity. Comments are listed below:

1. It would be useful if at the end of the introduction there were bulleted research objectives, and they will be discussed only in the discussion section.

2. There is no information about what statistical tools were used to analyze and display genetic data. It is a critical shortcoming If Student's t-test was used only.

3. It would be good to present the positions of the species found on maps against the background of their Euro-Asian ranges.

4. The finding of ‘unknown Beta sample’ is very interesting and this section should be developed. Based on morphological and genetic analyzes, it should include the full Latin name of the new taxon in the rank of species, subspecies or varietas. The information and name of the newly described taxon should also be included in the abstract and keywords.

5. Figures generated with the use of advanced numerical tools are missing.

6. The discussion should also discuss other non-climatic factors that may affect the differentiation of the studied species.

7. At the end, a few compact conclusions resulting from the results obtained should be added.

Author Response

Rev-plants-1873196

The article entitled ‘Distribution of Biodiversity of wild beet species (genus Beta L.) in Armenia under ongoing climate change conditions’ is interesting and may be published in Plants with major corrections. The article deals with important issues related to the protection of centers origin of Beta genus diversity. Comments are listed below:

  1. It would be useful if at the end of the introduction there were bulleted research objectives, and they will be discussed only in the discussion section.

We have now revised the last paragraph of the introduction according to the suggestion of the reviewer (lines 138-144).

  1. There is no information about what statistical tools were used to analyze and display genetic data. It is a critical shortcoming If Student's t-test was used only.

For the analysis of the genetic data we implemented multiple sequence alignments using Clustal Omega and for representation of the phylogenetic relation in the phylogenetic tree we used the maximum likelihood (ML) algorithm as a statistical method, using Molecular Evolutionary Genetics Analysis (Mega X) (Section “4.4. Sequence Alignment and Phylogenetic Analysis”, Lines 453-455 and 457-460 of the original manuscript). These analyses are now described more detailed in lines 254-257 and 535 - 542.  

The student’s t-test was used for analyzing the small dataset for morphological differences of the samples, as it was specified in the text. Mentioning the student t-test in a separate paragraph “4.3. Statistical analyses” could have been misleading, so we have now removed it, and added the statistical analysis of the morphology data into the paragraph “4.2. Morphological Study”. (lines 500-501).

  1. It would be good to present the positions of the species found on maps against the background of their Euro-Asian ranges.

The additions are made as suggested by the reviewer. The distribution range of the species is added to the supplementary material (Figure S3).

  1. The finding of ‘unknown Beta sample’ is very interesting and this section should be developed. Based on morphological and genetic analyzes, it should include the full Latin name of the new taxon in the rank of species, subspecies or varietas. The information and name of the newly described taxon should also be included in the abstract and keywords.

We have now revised the section about the “Unknown Beta”, as suggested by the reviewer (lines 267-270). The full Latin name is suggested and included in the abstract (line 23) and keywords (line 30).

  1. Figures generated with the use of advanced numerical tools are missing.

We have now modified Figures S1-3. We are not quite sure what kind of modifications are suggested by the reviewer, and we are ready to make further modifications if needed.

  1. The discussion should also discuss other non-climatic factors that may affect the differentiation of the studied species.

We have now discussed other non-climatic factors that may affect the differentiation of the studied species (lines 367-370 and 417-439).

  1. At the end, a few compact conclusions resulting from the results obtained should be added.

We have now revised the conclusion as suggested by the reviewer (lines 556-563).

Reviewer 2 Report (New Reviewer)

I found this an interesting and potentially useful contribution, though I did worry a little about using as citations the secondary national reports to the Climate Change Convention and the Convention on Biological Diversity to be a bit dubious. You will see my detailed comments attached.

Author Response

Reviewed by Prof. Jeffrey A. McNeely

Note that suggested wording changes are underlined.

  1. I have not yet had the opportunity to visit Armenia, but having worked on agroecosystems, biodiversity, and climate change thus gaining experience in many other parts of the world, I was intrigued to read this article that clearly linked the biodiversity of wild beet species with climate changes. It made a compelling case about the very real potential of climate change affecting the distribution, and perhaps evolution, of genus Beta, an important food, forage, and medicinal plant. Such a finding may also be relevant for other domesticated plants growing in habitats that are vulnerable to changes in the climate variables that are subject to change as the drivers of climate change continue to strengthen.

  1. Any paper that includes “biodiversity” in its title should explicitly remind readers that it is using the definition of the term contained in the Convention on Biological Diversity. This definition essentially includes the variability of genes, species, and ecosystems, so the article should also include these three elements.  This submission is very strong of the first two but is not as effective on the ecosystem dimension though it does describe the habitat changes that are expected under the current projections of climate change.  It would be helpful to include in the conclusions a recommendation on how the key ecosystems where Beta is found in Armenia can best be protected, perhaps by including them within Armenia’s three State Reserves or four National Parks.  It would even be worth exploring ways of conducting research in collaboration with Iran on Beta that may be found in its Arasbaran Biosphere Reserve that borders Armenia; such collaboration could well be of interest to Unesco as part of its Man and the Biosphere Program. 

We now added some dimensions of ecosystems in the manuscript, including keywords (line 29), introduction (lines 38-42 and 85-88), discussion (lines 371-392), conclusions (lines 561-563), and supplementary material (Figure S1) and came up with recommendations on protection of the ecosystems where Beta is found in Armenia (lines 570-597). The data on Beta species in the region is extremely scarce, and we would be happy to further explore the topic within the international collaboration with Iran.

  1. Line 38, better to say “…of which the number of endemic species is 144…”. (words underlined replace the word “where”).

Line 38 is now corrected according the reviewer’s suggestion (line 43).

  1. Since climate is a focus of the ms, more general relevant data are required. For example, l. 45 says Armenia has a dry climate, so say what is its annual rainfall and indicate its seasonality (details are in Figure 9, much later in the paper, but the larger picture could be painted here). And the sentences on lines 46-56 are a little awkward.  Consider instead “From 1929 to 2015, Armenia’s annual mean temperature increased by 1.23 C., and the annual average precipitation decreased by about 9% from 1935 to 2016.  Most of this decrease occurred in the past two decades, a time that was characterized by an increased frequency and intensity of extreme weather events and natural hazards like droughts, heat waves, hailstorms, and forest fires.”  And it would help if you could provide data supporting the claims of weather-caused alterations in phenological indicators, etc. in lines 52-54.

A separate paragraph about climate in Armenia is now added to the Introduction and supplementary material, as suggested by the reviewer (lines 49-65, Figure S2).

The lines 46-56 are now changed as suggested by the reviewer (lines 68-73).

Supporting literature on weather-caused alterations in phenological indicators is now added to the Introduction, as suggested by the reviewer (line 76).

  1. 56-65. These sentences raise some questions about using national communications to the Secretariats of the Climate Change Convention (213 pp in 2020 and 102 pp in 2010) and report to the Convention on Biological Diversity (165 pp in 2019) (dates and page numbers from citations 7,8,9). No question that these are important documents, but are they published and more widely available?  More important, these are national compilations that are secondary reports that draw on a wide range of primary research; so why not cite the primary research, assuming that the reports are properly referenced?  Here is a way this information could be presented, if it is acceptable to the Plants journal, here is a new line 45:  Armenia’s communication to the Convention on Climate Change has presented the best available evidence on the country’s climate change, reporting that “as a mountainous country with a dry climate (and continue with lines 46-56, but incorporating my comment in paragraph 4 above). Then, starting at line 56, say:  Armenia’s 2019 Sixth National Report to the Convention on Biological Diversity reported that “forest areas in Armenia have been significantly reduced, …..(9).” (include text from l. 57-61).

Yes, the reports are published, and we have now included the missing links to these reports in the References (lines 628-631, 650-651, 656-657, 729-730).

We found also some available primary research data, and included them as a reference, where it was appropriate (line 88).

We rephrased the citation of the reports according to the suggestion of the reviewer (lines 66-68, and 76-81).

  1. The mention of the use of beets in traditional medicine (line 72) would be stronger if it indicated the kinds of afflictions it was used to treat.

More specific information about use of beets is now included in the Introduction, as the reviewer suggested (lines 97-99).

  1. Line 89, would it be better to say, “…five have been recorded in the territory of the Republic of Armenia.” Not “mentioned”, which has no clear meaning.

Line 89 is now corrected as suggested by the reviewer (line 117).

  1. L. 94. What does “most annotated species” mean?  Most often revised?  Most often cited?  Or…?

Line 94 is now corrected according to the suggestion (line 122).

  1. l. 105. Better to say, “…has been prepared.” Or compiled.  Not “shown.”

Line 105 is now corrected according to the suggestion (line 134).

  1. l. 110. Better to say, “recovery of genetic resources that are endangered or have been lost and may have become extinct.  Not “destructed”

Line 110 is now corrected according the reviewer’s suggestion (lines 143-144).

  1. l. 120, better to say, “Over the past two decades our group has sent numerous expeditions to record the growth and distribution of the wild beets.” Are these really “expeditions”, or rather “fieldwork in Beta habitats”?

Line 120: The correction is now made according to the reviewer’s suggestion (line 153).

  1. l. 132. “altitudes of 1675 m a.s.l. (Table 1)” needs to say “altitudes from 1400 to 1980 m a.s.l (Table 1).” According to my reading of Table 1.  Is this correct?  Giving a single altitude is misleading when the beets occur over a wide elevation range. 

We meant here that the species in the selected location was preserved at the same altitude range during 2016-2021. We now removed the annotation of Table 1 to avoid any misleading.

  1. l. 133. Better to say, “…has decreased in recent years, as they are preserved….”  Not “during the last years”

Line 133 is now corrected according the reviewer’s suggestion (line 167).

  1. l. 149, better to say, “…distribution has declined over the years….” Not “reduced”.

Line 149 is now corrected according the reviewer’s suggestion (line 183).

  1. l.150, better to say, “….around Lake Sevan, where the species was reported from the 1980s to early 2000 (Table 1)”. And l. 152, say “…earlier studies were still present (Table 1). Not “preserved.”

Lines 150 and 152 are now corrected according the reviewer’s suggestion (lines 184-185 and 186).

  1. l. 159, say “…were collected as herbarium specimens and registered….”

Line 159 is now corrected according the reviewer’s suggestion (line 193).

  1. l. 193, say “..B. macrorhiza earlier described in that region.” Not “before”.

Line 193 is now corrected according the reviewer’s suggestion (line 227).

  1. l. 194. Is “Unknown Beta” a new species?  Might need to be at least suggested as a possibility. And on l.209, “Unknown Beta did not fully fit the identification features of any of the described wild beet species (Table 3), support for it being a new species, or at least genotype.  And l. 212, delete “there” (not needed here; the point is clear without it).

The possibility of the “Unknown Beta” to be a new species is now suggested, as advised by the reviewer. A new Latin name is provided for the proposed species. Yet, further studies would be needed for correct identification of the rank of the taxon (lines 267-270).  

The corrections in lines 209 and 212 are done as suggested by the reviewer (lines 243-244 and 246, respectively).

  1. l.214. Say, “We also included….”

Line 214 is now corrected according the reviewer’s suggestion (line 248).

  1. l. 216, say, “…region, though at a lower altitude,…”. Not “on a lower altitude”.

Line 216 is now corrected according the reviewer’s suggestion (line 250).

  1. l. 228, better to say, “We could suggest that the collected plant is a hybrid…”. Not “found plant”.

Line 228 is now corrected according the reviewer’s suggestion (line 263).

  1. l. 245. “…was the closest to the…”. (spelling correction)

Line 245 is now corrected according the reviewer’s suggestion (line 284).

  1. l. 251-2, say, “The species grows in dry climate….most prone to the impacts of climate change.

Lines 251-252 are now corrected according the reviewer’s suggestion (lines 291-292).

  1. l.257-8, say, “…annual precipitation was highly variable between 2010 and 2020, with the 2013 rainfall less than half that reported in 2010 but increased to previous figures by 2016 before again declining to the 2013 rainfall level by 2020 (Figure 8A). Along with temperature changes, these climatic factors likely affected plant growth.

Lines 257-258 are now corrected according the reviewer’s suggestion (lines 297-300).

  1. l. 270, delete “most probably” and instead say, “Nevertheless, the climate factors are likely to be the most important factors affecting the observed effects.”

Line 270 is now corrected according the reviewer’s suggestion (lines 309-310).

  1. l. 279, “…natural habitats, as they were likely dispersed to the lower areas by humans.” Not “most probably”

Line 279 is now corrected according the reviewer’s suggestion (line 318).

  1. l. 285. “The gradual temperature increase of around 1.7 C….

Line 285 is now corrected according the reviewer’s suggestion (line 324-325).

  1. l. 292. Say, “We also suggest that the changes in the distribution of B. corolliflora during the past several years were not associated only with the changing climate. “

Line 292 is now corrected according the reviewer’s suggestion (line 331-333).

  1. l. 294. “….heavily exploited zone with intensive agriculture and increased urbanization, so the involvement of the anthropogenic factor should be seriously considered [33, 34]”.

Line 294 is now corrected according the reviewer’s suggestion (line 334).

  1. l.310, delete “the” so it reads, “…could be related to climate change.”

Line 310 is now corrected according the reviewer’s suggestion (Line 348).

  1. l. 313, say, “…average annual precipitation has gradually decreased since 2010, then increased in 2016 and 2018 (Figure 8C).”

Line 313 is now corrected according the reviewer’s suggestion (lines 351-352).

  1. l. 316. Is it “species migration”, or “dispersal”?

Species “dispersal” is a better fit. Line 316 is now corrected according the reviewer’s suggestion (line 354).

  1. l. 331. described (a spelling correction)

Line 331 is now corrected according the reviewer’s suggestion (line 394).

  1. l. 337-8. Say, “Further studies with a larger sampling….deeper assessment of phylogenetic relations and gene flow between the species.”

Lines 337-338 are now corrected according the reviewer’s suggestion (lines 400-401).

  1. l.375-379 seems more appropriate to be in the conclusions and recommendations. The points are good, but “legislation improvement” is too general; it would be better to specify what should be included in the legislation, or call for reviewing the relevant national legislation by a committee that includes expertise in Armenia’s plant genetic resources. And this raises the question of whether this has been included in Armenia’s report to the Convention on Biological Diversity and its report on Climate Change (cite 9, 7, 8).  If so, why not?  And what needs to be done?  This article has provided the data to support such a recommendation to the appropriate government agencies in Armenia.

The lines 375-379 are moved to the Conclusion and Recommendation section (lines 578-580), where suggestions for preservation of the species and ecosystems is given. 

  1. l. 396. Better to say, “…population size was determined by counting the individuals…”. Not “estimated” if you are actually counting the individuals, but a much more accurate datum.

Line 396 is now corrected according the reviewer’s suggestion (line 478).

  1. l. 401-2. Say “…the expeditions were also organized in September-October for seed collection…”. Or is it more accurate to say, “…the fieldwork was also organized in September-October for seed collection…”

Lines 401-402 is now corrected according the reviewer’s suggestion (line 484).

  1. l. 407-8. Consider:  “In addition to the mentioned locations, seasonal fieldwork by our group to different natural habitats of the wild beets has continued since 2001, collecting species distribution data and developing reliable maps of the occurrence of the various Beta species in Armenia.”   

Line 407-408 is now corrected according the reviewer’s suggestion (lines 490-493).

  1. Section 4.4, on DNA extraction, etc. seemed solid, but I claim no detailed expertise in this field.

  1. l. 464. Say, “Climate change affecting Armenia varies substantially from region to region, as indicated by our floristic zones.

Line 464 is now corrected according the reviewer’s suggestion (line 546).

  1. Section 5, conclusions, well reflects the genes, species, and ecosystems package of the Convention on Biological Diversity. Above, I have suggested some ways that this section could be made more specific, in particular strengthening Armenia’s reports to the Biodiversity and Climate Change conventions, and using these to strengthen the country’s efforts to conserve the diversity of beet species and varieties (in other words, genetic diversity) as well as the habitats where the Beta is flourishing, and helping Armenian nature adapt to climate change.  And consider changing the last sentence slightly: “A combination of broader ecological and genetic data is providing a solid foundation for better understanding the phylogeny and distribution of Armenia’s wild beets as a basis for conserving this important element of the country’s natural heritage, and helping this biodiversity to contribute to the changing environmental conditions being driven by accelerating climate change.

Section 5 is now revised according to the suggestion of the reviewers (lines 545-597).

Round 2

Reviewer 1 Report (New Reviewer)

The authors responded satisfactorily to all comments and I recommend the article for publication.

Reviewer 2 Report (New Reviewer)

Already sent and comments were addressed satisfactorily 

This manuscript is a resubmission of an earlier submission. The following is a list of the peer review reports and author responses from that submission.

Round 1

Reviewer 1 Report

I think that this manuscript is unsuitable for Journal of Fungi.

Authors should choose related Special Issue 'Biodiversity, Distribution and Conservation of Plants and Fungi; Effects of Global Warming and Environmental Stress' in Plants.

Reviewer 2 Report

The article is well written, interesting, and contains important conclusions.
The Journal of Fungi publishes articles related to pathogenic fungi, fungal biology, 
and all other aspects of fungal research. The article does not contain any research on these topics.
The article is about the biodiversity of wild beet in Armenia.

Some changes should be introduced into the discussion and, for example, 
the description of precipitation and temperature should be added to Materials and Methods.
I couldn't find the results of the analysis of variance, were there homogeneous groups?

Perhaps a better suited journal to consider for authors would be Sustainability.

Reviewer 3 Report

I commend the authors of the manuscript titled “Distribution of Biodiversity of Wild Beet Species (Genus Beta 2 L.) in Armenia under Ongoing Climate Change Conditions” for their work on the geographical distribution and morphology of wild beet species in Armenia,.

Before this manuscript is published, there are several things need to be addressed or corrected:

  • In the abstract, the abstract is very short and need to include more results.
  • In the introduction:
  • The introduction is comprehensive and no changes need to be done there

  • In the results
  • Table 3 need to be converted to figure.
  • The photos of the roots need to be shown if available as well as leaves and flowers
  • Figure 6 need to be sharpened and enlagrged
  • In the discussion part: table 6 need to be converted to figures.

Also the discussion is very long and need to be shortened

  • In the material and methods:

 section 2.1 :  why only these three locations were selected? Should be explained

  • The study would be much comprehensive and more economical and have applicability if data included chemical composition of the roots.

Section 2.4, why these 2 regions were used? trnL-trnF intergenic spacer (LF) and the partially sequenced alcohol dehydrogenase (adh) of nuclear DNA. There are dozens of barcodes? Was there pilot study before? Or they were successful on the same family on other species or same species?

  • The conclusion is long and need to be shortened.

Reviewer 4 Report

The paper is interesting and contains valuable knowledge, both in terms of geographical region and plant species. However, it doesn't fit the scope of the journal entitled "Journal of Fungi". Authors have selected the journal focused on one certain taxonomic group of organisms (Kingdom Fungi), but the paper contains no data on fungi.

So please, modify the research design and methods, and add the fungi to the study. For example, mutualistic interaction between beet species and mycorrhizal fungi and/or antagonistic interactions of beets with pathogenic fungi would be interesting. In the context of ongoing climate change, plant-fungal interactions seem to be particularly important.

Otherwise, please consider other journals such as Plants (https://www.mdpi.com/journal/plants ) or Agriculture (https://www.mdpi.com/journal/agriculture). It's a better choice for the study of plant species only.

In the presented form, the paper cannot be accepted in the Journal of Fungi.